# *Theobroma cacao* and *Theobroma grandiflorum*: Botany, Composition and Pharmacological Activities of Pods and Seeds

**DOI:** 10.3390/foods11243966

**Published:** 2022-12-08

**Authors:** Elodie Jean-Marie, Weiwen Jiang, Didier Bereau, Jean-Charles Robinson

**Affiliations:** Laboratoire COVAPAM, UMR Qualisud, Université de Guyane, 97300 Cayenne, France

**Keywords:** cocoa, cupuassu, pods, beans, pharmacology, nutrition, bioactive compound

## Abstract

Cocoa and cupuassu are evergreen Amazonian trees belonging to the genus *Theobroma*, with morphologically distinct fruits, including pods and beans. These beans are generally used for agri-food and cosmetics and have high fat and carbohydrates contents. The beans also contain interesting bioactive compounds, among which are polyphenols and methylxanthines thought to be responsible for various health benefits such as protective abilities against cardiovascular and neurodegenerative disorders and other metabolic disorders such as obesity and diabetes. Although these pods represent 50–80% of the whole fruit and provide a rich source of proteins, they are regularly eliminated during the cocoa and cupuassu transformation process. The purpose of this work is to provide an overview of recent research on cocoa and cupuassu pods and beans, with emphasis on their chemical composition, bioavailability, and pharmacological properties. According to the literature, pods and beans from cocoa and cupuassu are promising ecological and healthy resources.

## 1. Introduction

Cocoa and cupuassu are two Amazonian fruit trees from the genus *Theobroma* and the family Malvaceae. They have a lot of vernacular names, including Cacaoquahuatl (Aztec), Cacau-Da-Mata (Brazil), Cacauí/Cupui (Portuguese), Coklat (Indonesia), Copoasu/Cupuaçuzeiro (Brazil), Cupuaçú (Portuguese), Bacau (Colombia), Copuazú (Spanish) or Lupu (Suriname) [1]. Cocoa (*Theobroma cacao* L., 1753) is native to Central (Mexico) and South America (From French Guiana to Bolivia), while cupuassu (*Theobroma grandiflorum (Willd. ex Sprengel) K. Schumann*) is from the northeastern states of Brazil (Pará and Maranhão) [2,3]. These species require specific pedoclimatic conditions related to temperature (18–32 °C), rainfall (1250–3000 mm), and soil pH (6–7.5) [1,4]. Though the trees morphologies are relatively similar, their flowers and fruits differ in shape and color. The shape of cocoa pods varies from warty to almost smooth, ranging from bright green to dark green, yellow to dark red, or a combination of all these colors [5]. Cupuassu ranges from light brown to dark brown and can be oblong to ellipsoidal [1].

Each part of the plant has its own qualitative and quantitative compositions. Fruits (pods and beans) are sources of protein, fat, and carbohydrates, but also polyphenols and methylxanthines [1]. Although cocoa beans are widely studied because of their involvement in the agri-food sectors, its pod, considered a waste, is studied little. Indeed, its chemical composition remains largely unknown even if it suggests some biological potential. Studies highlighted the presence of secondary plant metabolites in these fruits that are well known to be responsible for certain human health benefits. In many studies, cocoa polyphenols are involved in protecting against cardiovascular, tumor, oxidizing, and neurodegenerative damage [6,7,8].

Furthermore, studies on cupuassu fruits have shown a growing interest in the development of by-products in agri-food and cosmetics. However, data on the composition and health potential of its fruits remain very rare. Consequently, this article aims to provide an overview of recent data on cocoa and cupuassu regarding taxonomy, agroecology, phytochemistry, and biological properties. Moreover, it aims to highlight the roles of their different vegetative organs.

## 2. Botany of Cacao and Cupuassu

### 2.1. Taxonomy and Origins

Cupuassu (*Theobroma grandiflorum (Willd. ex Sprengel) K. Schumann.)* is native to Brazil in the northeast of Maranhão and the south/ southeast region of the state of Para (covering the central regions of the Tapajós, Tocantins, Guamá, Xingù, and Anapú rivers) [2]. It can also be found in the Guianas shield (French Guiana, Guyana, Surinam, Colombia, Venezuela), Ecuador, and Costa Rica [1,9]. Cocoa (*Theobroma cacao* L., 1753) is native to Central (such as Mexico) and South America (from French Guiana to Bolivia) [3,10]. Some genetic groups are thought to be associated with specific regions. For example, Amelonado and Guiana could have an eastern Amazonian origin while Criollo and Nacional could have a western Ecuadorian Amazonian origin [11,12].

In 1882, Morris divided cocoa into two groups: Criollo and Forastero (with different Forastero varieties named Cundeamor verugoso, Liso, Amelonado and Calabacillo) [13]. In 1886, floral morphology described by Schuman et coll., revealed a difference between *T. cacao* and *T. grandiflorum* [2]. In 1892, Hart gave a new ranking based on three groups with different varieties: Forastero (Amelonado, Ordinary and Cundeamor), Calabacillo (Colorado and Amarillo), and Criollo (Colorado and Amarillo) [14]. In 1901, Preuss identified a new group called Trinitario and considered that cocoa could be classified into three varieties: Forastero, Criollo, and Trinitario [2]. The emergence of Trinitario could be explained by the spontaneous hybridization between Criollo (already cultivated) and Forastero (introduced), in Trinidad in 1727 [15,16]. In 1964, Cuatrecasas defined that the genus *Theobroma* would be divided into six sections containing 22 species [2], which will be presented in Table 1. 

The classification of cocoa has long been based on morphological criteria and this distinction of Forastero, Criollo, and Trinitario is still used today. However, since 2008, a new classification based on genetic criteria has been demonstrated and 10 genetic groups have been defined: Maranon, Curaray, Criollo, Iquitos, Nanay, Contamana, Amelonado, Purus, Nacional, and Guiana [11], suggesting that the morphogeographic classification is now obsolete. For cupuassu, few data are available on taxonomic distinction. In Brazil, three cultivars of cupuassu are well known: Redondo (with a rounded apex), Mamorano (with a pointed apex), and Mamau (a hypotetic parthenocarpic mutant) [1].

### 2.2. Agroecology and Geographical Distribution

Many pedoclimatic parameters can modulate the developments of cocoa and cupuassu trees. Both require croplands where the average temperature is 25 °C (18–32 °C for cocoa and 21.6–27.5 °C for cupuassu) [1,4]. Inadequate temperatures could threaten vegetative growth, flowering, and the maturity of fruits [17]. Cocoa is sensitive to prolonged exposure to the sun and the wind; therefore, it should evolve in the protective shade of trees in its surroundings [1,18]. They occur in very similar wetlands (70–88%) and require an average rainfall of 1250–3000 mm for cocoa and 1900–3000 mm for cupuassu [1]. There was a significant relationship between pre-harvest precipitation (2–5 months) and bean weight in Nigeria [19]. Excessive precipitation or long periods of drought can harm the plant by increasing the risk of fungal diseases [10]. The distribution of various varieties is shown in Figure 1.

Cocoa grows in rich, deep, and well-drained soil. It must be composed of 50% sand (large particles induce good drainage); 30–40% clay (small particles for water and nutrient retention) and a balance of 10–20% of medium-sized silt. For cocoa cultivation, the appropriate soil types would be entisols (fresh horizontal alluvial soils with minerals), inceptisols (of volcanic or other origin with minerals and low horizon development), and red or yellowish ultisols and alfisols (minerals-rich forest soils with intense leaching) [1,23]. With very close conditions, both species can be found in the same growing areas. Cupuassu may also grow in red oxisols and inceptisol yellow-red oxisols [25,26]. pH is also an important parameter since although cocoa can grow at a pH of 4 to 8.5, its optimal pH is between 6 and 7.5 where nutrients and trace elements are more available [1,23]. This availability is influenced by soil pH [27] because at pH 7.5 and 8.5, phosphorus, manganese and boron may be less available [28], while acidic soil induces low levels of phosphorus and high levels of iron and copper salts, known to have toxic effects [23]. Alfaia et coll., suggested that cupuassu could growth in a soil pH range from 4 to 5 [25].

### 2.3. Plant Morphologies

Cocoa and cupuassu are evergreen and cauliflory trees. Cocoa is usually 4–8 m tall, and rarely grows up to 20 m, with dark grey-brown bark, whilst cupuassu can grow 5–15 m tall and has a brown bark trunk [1,2]. Morphologically close, their leaves are simple and alterne, ranging from narrowly ovate to obovate-elliptic. The veins pinnate towards the extremity with a well-marked main vein and prominent secondary veins. They range from 20 to 35 cm long and 6 to 10 cm wide [1,29]. 

Flowers of cocoa (white, yellow, or pale pink) and cupuassu (white to yellow, often with red lines) can be found on the trunk or branches as a “floral cushion” [2,29]. The flowers are bisexual, pentamerous, and have a calyx with five lobes divided almost at the base. The five stamens are curved outwards, and the anthers alternate with 5 staminodes (usually purple), and a pentagonal upper ovary with five loculas containing numerous ova [1,29]. Some varieties may have specificities. Indeed, four floral quantitative descriptors (sepal width, gynoecium length, number of ovules, and particularly the width of the petal ligule) could be used to distinguish the Guiana from the Trinitario, Forastero, and Amelonado populations [30]. Fertilization (allogamic or heterogamic) causes an increase in the ovary which becomes a fruit called «cherelle» (during the growth phase) then «pod» (at the final stage). This maturation varies depending upon the variety and lasts about 4–7 months for cocoa [29,31], and 3–6 months for cupuassu [32]. Both species face fertility challenges, resulting in low fructification rates. The known reasons may be (1) lower rates of pollinated flowers, (2) lower effectiveness of entomophilic pollination, and (3) incompatibility reactions inducing flower abortion [29,32].

Indeed, Cuatrecasas et coll., pointed out that the problem of incompatibility is one of the specificities of the *Theobroma* species [2]. Cupuassu and cocoa have both been identified as self-incompatible species [32]. Self-incompatibility and cross-incompatibility are defined by the inability to pollinate the flower (by its own pollen or by the pollen from another incompatible tree) to turn it into fruits. These reactions occur late after a normal increase in the pollen tube and can be induced (1) by the absence of antheroids in the embryonic sac, (2) by the absence of gamete fusion, or (3) by the abortion of the egg after fusion. Furthermore, pollinators are not very effective due to the natural floral barriers [29,33,34]. These incompatibility reactions are therefore influenced by gametophytes alleles and their dominant position in the parent plant [35]. Cope et coll. hypothesized that incompatibility would be based on the expression of two genes A and B (coding for dominant alleles A and B and recessive alleles a and b) which could affect the expression of an S gene (highly incompatible factor). Self-incompatibility would be induced by the presence of a single dominant allele (A or B), for an allele that carries the same allele S. On the other hand, compatibility would be possible when one of the two genes A and B are in a recessive homozygous form since they would result in an inactive S system [36]. However, the latter S system (controlled by five alleles) would have dominant and co-dominant relationships and if a S allele is in a dominant form in a parent, incompatibility would be maintained [37]. With ovaries collected after 72 h of pollination, Ramos et coll., showed that compatible- and incompatible-crosses induced a small percentage of ovules with no indication of gamete fusion, but at a higher rate for incompatible-crosses. They suggested that fusion could occur, but that two factors could act at the same time: the delay in incompatible pollen tube growth and the ovular inhibition before fertilization [33]. 

Although cocoa and cupuassu fruits have very distinct shapes and colors, the fruits are oblong, obovate, or sub-globose. Shape, weight, and color may vary depending on the variety and the pedoclimatic conditions. The ripe cocoa pod is 10–32 cm long and 6–15 cm wide. The shape can be spherical to cylindrical, and the surface (with ten longitudinal grooves) can be warty and deeply pleated to almost smooth. The tip, called the apex, can be pointed or serrated. Color varies during ripening from bright green to dark green, yellow to dark red, or a mixture of all these colors [5]. The ripe cupuassu pod is 20–25 cm long and 6–10 cm wide. It can be oblong to ellipsoidal, woody, pubescent, and comes in shades of brown [1]. Pod shapes are given in Figure 2.

In general, pods contain about 20 to 60 beans in five rows (corresponding to the 5 lobes of the ovule), surrounded by a sweet mucilaginous pulp [1,5,18]. For cocoa, beans are ovoid, ellipsoid, amygdaloid, more or less complanate, or round, ranging from 20–40 mm long and 12–20 mm wide with a white or light-yellow wrapped pulp. For cupuassu, seeds are ovoid or ellipsoid-ovoid, more or less flattened, and vary from 20–30 mm in length and 20–25 mm in width with a yellowish pulp [2]. Color, height, and weight may vary depending on genetic and pedoclimatic factors. For cocoa, Criollo has light-colored beans while Forastero may have white to dark-purple beans and cupuassu has a light brown-color [2]. Beans are composed of two parts: the almond (two cotyledons and an embryo) and the shell (thin film that protects the beans). Early in germination, cotyledons induce photosynthesis and provide the nutrients required for embryo development through an enzymatic breakdown of the reserves [41].

## 3. Chemical Composition of Beans and Pods

### 3.1. Macronutrients 

Pods (pericarp) and beans (pulp and seed) account for 80% and 20% of the total fruit weight of cocoa, respectively, and for 43% and 57% in cupuassu [42,43]. Each vegetative part has its own macronutrient composition that includes proteins, lipids, and sugars, essential to its metabolism. For example, in cocoa, protein function is dedicated to 48% for metabolism and energy, 13% for protein synthesis, and approximatively 8% and 7% for membrane transport and defense, respectively [44].

#### 3.1.1. Proteins

The protein content of cocoa pods varies from 2.4 to 17.6 g/100 g of the Dried Weight [45,46,47,48,49,50]. Cocoa pods’ (COPs) protein content (from Africa, Brazil, Ecuador, Colombia, and Guinea) averaged 15 ± 1.4%, of which fraction albumin + globulin was for 11.3 ± 1.1%, fraction glutenin for 2.5 ± 0.3%, and fraction prolamin for 0.4 ± 0.09%, respectively. Glutamic and aspartic acids were the major amino acid (respectively 1.9 ± 0.18% and 1.5 ± 0.12% of the COPs weight). With a total amino acid content representing 11.6 ± 0.9% of COPs, we can find lysine, leucine, threonine, and valine averaging at 0.8%; isoleucine, tyrosine and phenylalanine at 0.5%; cysteine and histidine at 0.25%; tryptophane at 0.12%; and methionine at 0.06%, respectively. They represent essential amino acids and count for 44.6% of the total amino acid content [45]. To our knowledge, no data is available concerning cupuassu pods (CUPs).

These contents range from 2.5 to 14.4 and from 2.0 to 26.2 g/100 g of DW in cocoa beans (COBs) and cupuassu beans (CUBs), respectively [51,52,53,54,55,56,57,58]. These include functional proteins such as vicilin (reserve protein and precursor of specific aroma) and trypsin inhibitors (related to seed germination and fungal defense) [53]. Isolated from 20 g of cupuassu beans, albumin, globulin, prolamin, and glutelin accounted for 3.5 ± 0.1, 0.7 ± 0.1, 2.8 ± 0.8, and 1.5 ± 0.2% of protein extracted of each fraction, respectively [56]. With a total protein content of 26.2 ± 0.3% of the dry weight of beans, Carvalho et coll., indicated that leucine, valine, and threonine represented the major essential amino acids (8.44 ± 0.15, 7.26 ± 0.15, and 5.92 ± 0.04 g/100 g of protein, respectively) and Glutamic and Aspartic acids the major non-essential amino acids (14.83 ± 0.11 and 12.38 ± 0.26 g/100 g protein, respectively) [56].

#### 3.1.2. Lipids

COPs have low lipid contents, ranging from 0.6 ± 0.2 to 2.3 ± 0.4 g/100 g of DW [45,46,48,49,50]. Essential human fatty acids such as palmitic, stearic, arachidic, and linoleic acids, and also pentadecanoic acid (that is a rare saturated fatty acid in nature), have been reported [59]. As far as we know, there is no data regarding CUP’s lipid contents.

Concerning beans, lipid levels are higher than in the pods and account for 20 to 60% of their weight [43,52,54,55]. COB’s lipid composition includes nearly 97% of glycerolipids, mainly triglycerides such as 1,3-dipalmitoyl-2-oleyl-glycerol (16.4–21.4%), 1,3-distearoyl-2-oleoyl-glycerol (22.8–32.9%), and 1-palmitoyl-2-oleoyl-3-steraoyl-glycerol (38.0–46.2%), but also 3% of glycolipids and unsaponifiable matter [55,60,61]. Fatty acids are significant in lipid composition including the predominant saturated fatty acids, stearic acid (28.9–39.4%) and palmitic acid (27.2–32.9%). Oleic acid is about 27.4 to 37.9% of the unsaturated fatty acids [55,61,62]. For CUBs, oleic acid (36.3–42.2%), stearic acid (29.2–32.9%), arachidic acid (9.8–11.2%), and palmitic acid (7.3–7.8%) are the major fatty acids [54,63,64]. Comparing CUBs and COBs respectively, palmitic acid level is 3 times lower, arachidic acid level 10 times higher, and stearic acid level equivalent [64]. In terms of the saturated/unsaturated fatty acids balance, COBs is composed of 69.3% and 30.7% while CUB is 48.9% and 51.1% for each group, respectively [65].

#### 3.1.3. Carbohydrates

In COPs, total sugars account for 1.7 ± 0.3 g/100 g of DW, where glucose is the main sugar (1.1 ± 0.2%), followed by fructose (0.6 ± 0.2%), and sucrose (noted as trace). The total starch stands for 1.1 ± 0.2% [45]. To our knowledge, no data is available for CUPs. 

For cocoa and cupuassu beans (COBs and CUBs), total sugars range from 0.1 to 3.1 g/100 g of DW and from 1.3 to 1.6 g/100 g of DW, respectively [66,67]. Total carbohydrate accounts for 10.5–19.0 g/100 g of DW for COBs, and 13.6–23.1 g/100 g of DW for CUBs, respectively [43,51,54]. In COBs, glucose (29.0–127.3 mg/100 g of DW), fructose (12.3–69.1 mg/100 g of DW), and sucrose (265.0–2887.0 mg/100 g of DW) were measured. Other sugars such as melibiose (4.9–197.6 mg/100 g of DW) and myo-inositol (30.8–85.4 mg/100 g of DW) were also found in cocoa [67].

### 3.2. Micronutrients

Cocoa and cupuassu are well known as rich sources of polyphenols and methylxanthines [1,68]. These compounds are secondary metabolites located in the storage cells that are distinctive with their single large vacuole [57]. In beans, these cells are called “pigment-cells”, because of the existing correlation between their color (from white to dark purple) and anthocyanin contents. 

#### 3.2.1. Polyphenols

Polyphenols that are present in pods and beans from cocoa and cupuassu were identified by HPLC and listed in Figure 3.

An approximative total polyphenol content (TPC) using the Folin-Ciocalteu method showed that COPs had lower TPC than COBs: 3.2 ± 0.3–56.5 ± 0.6 mg gallic acid equivalent/g of DW) [46,69,77,78]) vs. 9.8 ± 0.1–202.2 ± 6.5 mg GAE/g of DW [75,79,80]. Polyphenol contents may vary according to various parameters such as genotype [81,82], maturity [83], soil, and crop conditions (sun exposure, number of fruits on tree, soil type) [82,84]. Pico-Hernández et coll., showed the importance of genotype. Indeed, compared to other Colombian cocoa clones (such as CCN-51), ICS-39 have higher TPC than the others [58]. Focusing on one common variety (Trinitario), there were variations between fermented and dried cocoa beans from Venezuela, Dominican Republic, Ecuador, and Colombia with 10.4 ± 0.5, 19.6 ± 1.1, 25.4 ± 1.4, and 37.7 ± 2.2 mg GAE/g of DW [84], respectively. 

Whilst no data is available on cupuassu pods, cupuassu bean liquor (crushed fermented and roasted beans) has lower TPC (7.84 ± 0.54 mg GAE/g of DW) than cocoa (28.45 ± 2.45 mg GAE/g of DW). Moreover, total flavanol content (TFC) can be determined by two standard methods (1) with p-dimethylaminocinnamaldehyde (DMACA) and (2) butanol. Cocoa liquors had higher TFC than cupuassu (DMACA: 22.0 ± 1.85 and 8.70 ± 0.75 mg/g of DW and Butanol: 70.6 ± 2.5 and 19.5 ± 0.86 mg/g of DW, for cocoa and cupuassu respectively) [85]. 

#### 3.2.2. Methylxanthines

Theobromine, caffeine, and theophylline are methylxanthines/alkaloids which can be found in cocoa [86,87]. Their structures are described in Figure 4.

HPLC quantifications indicated that COBs have higher theobromine (0.19–7.66 g/100 g of DW) and caffeine (0.18–2.08 g/100 g of DW) contents than COPs with 0.002–0.4 and 0.002–0.004 g/100 g of DW values [1,51,83,88,89,90,91], respectively. 

In a comparison study involving beans, COBs have higher levels of theobromine (3.3 vs. 0.1 g/100 g of DW) and caffeine (0.56 vs. 0.05 g/100 g) than CUBs [66]. Although cocoa beans present a low content of theophylline (0.2–0.37 g/100 g of DW), it was not detected in cupuassu [1,68]. As polyphenols, methylxanthine contents also depend on genetic, pedoclimatic, and crop conditions. Indeed, maturity stage influences theobromine and caffeine contents, which increase by 40% from immature to fully ripe [83].

A brief comparison of the chemical compositions of beans and pods is presented in Table 2. 

## 4. Pharmacological Activities 

The use of polyphenols and methylxanthines has been associated with protection against cardiovascular or neurodegenerative damage and other metabolic disorders [7,8]. As has already been noted, cocoa and cupuassu have a very narrow composition that could be linked to health benefits. Although cocoa bean studies are predominant, the following section attempts to compile and discuss the health potentials of its pod, and compare them with the cupuassu vegetative parts.

### 4.1. Antioxidant (AO) Activity 

Polyphenols are well-known to be positively correlated with AO activity [92]. Indeed, with a higher TPC (611 mg of gallic acid equivalent/serving) than black and green tea (124 and 165 mg GAE/serving, respectively) and red wine (340 mg GAE/serving), cocoa also has higher AO activities in DPPH (2,2-Diphenyl-1-picrylhydrazyl) and ABTS (2,2′-azino-bis(3-éthylbenzothiazoline-6-sulphonique)) assays [93]. Cocoa and cupuassu beans (COBs and CUBs) had DPPH activities ranging from 24.0 ± 0.0 to 1370.1 ± 1.4 and 19.1 ± 2.3 to 1438.2 ± 13.0 μmol Trolox equivalent (TE)/g of DW, respectively [75,79,85,92,94]. COBs also had ORAC (Oxygen Radical Absorbance Capacity) activities from 303.0 ± 5.0 to 1097.0 ± 111.8 μmol TE/g of DW, while CUB’s range from 136.3 ± 1.8 to 713.0 ± 18 μmol TE/g of DW [52,54,75,85,95,96]. Even if a trend could be observed, biological variability does not exclude the possibility that one variety could be richer in polyphenols and theobromine than another. As mentioned, the chemical composition is related to genotypic, pedoclimatic, or plant growing conditions. However, the literature does not deny that these two species are of major importance because of their antioxidant potential in vitro but also in vivo.

Human endothelial vascular (EA. hy926) and hepatic (HepG2) cell lines were pretreated with cocoa extract, then the oxidative stress status was induced by the addition of terbutylhydroperoxide (t-BOOH). Cocoa flavanol extracts, especially epicatechin, showed an effective ability to protect cells in (1) reducing Reactive Oxygen Species (ROS) generation, (2) reducing malondialdehyde (MDA) level, a lipid peroxidation marker, and (3) enhancing the activity of AO enzymes such as glutathione peroxidase (GPx) and gluthatione reductase (GR). Martin et coll., also showed the possible contribution of theobromine in the protective potential of cocoa extract against ROS generation [97,98]. In addition, human intestinal epithelial Caco-2 cells and murine enteroendocrine STC-1 cells (also stimulated by t-BOOH) showed that pretreatment with cupuassu extract was successful in reducing ROS production by approximatively 20% and 30% for Caco-2 and STC-1, respectively. Authors also indicated that in the in vivo rat model, cupuassu extract prevented the expected increase in ROS levels caused by food administration [99]. 

As mentioned, each part of plant contains its own polyphenols content, which would explain the variability of their AO activities. For example, cupuassu beans (CUBs) have a TPC and DPPH activity 80% higher than its pulp [52]. COPs have lower AO activities than COBs with DPPH values ranging from 18.4 ± 0.3 to 133.0 ± 1.0 μmol TE/g of DW and ORAC values from 25.1 ± 1.5 to 342.9 ± 3.5 μmol TE/g of DW [46,50,88]. By comparing COP’s and COB’s ABTS values, this trend is also observable with 23.1 ± 0.15–229.6 ± 21.7 vs. 925.5 ± 9.9–2610.0 ± 30.0 μmol TE/g of DW, respectively [46,50,79,88,94]. This trend could be explained by the difference in their polyphenol composition (Figure 3). Pico-Hernández et coll, mentioned that polyphenol fractions (monomers, dimers, trimers) may participate in a different way with the TPC and AO values of cocoa extract [58]. The AO potential of COPs is less studied, however, it could contribute a greater value to this waste and thus allow the creation of different ways of recovery. In fact, cocoa and cupuassu beans and pulp are mostly used in agri-food and cosmetic sectors. Each ton of dry beans results in ten tons of cocoa pods and becomes a huge organic mass to be eliminated because they can give rise to health problems (putrefying, fungal diseases) [100]. 

Some authors are still looking to add value to them and thus highlight notions of upcycling and the valorization of the total fruit. In an in vivo animal model, COPs were added in the diet of ewes for 8 weeks and their blood and milk AO statuses were analyzed. Authors found that COPs did not affect AO plasmatic capacities (FRAP and ABTS), but did increase superoxide dismutase (SOD, an AO enzyme) activity. Moreover, a COPs diet reduced the protein carbonyl level, a marker of oxidative protein damages, and increased SOD plasmatic activity [101]. This behavior seemed to be due to the presence of polyphenols, which are able to form complex plasma proteins through a binding affinity [102]. 

Process would affect the health potential of cocoa and cupuassu. Cocoa is mainly consumed in agri-food sectors as the basic ingredient in chocolate. Cupuassu can also be transformed into a chocolate derivative named “cupulate” [103]. A key step to switching from the bean to the derived chocolate (tablet or powder) is “fermentation”, which reduces the astringency of the beans and increases the characteristic taste and color of chocolate [4]. This process leads to composition changes, including a reduction of methylxanthine and polyphenol contents. Indeed, methylxanthines undergo external diffusion from cotyledon to the shell. Anthocyanidins are hydrolyzed to cyanidins, and sugar and flavonoids are converted to quinones, that can be complexed with polyphenols, proteins, or other compounds [4,104,105]. Decreases in polyphenols and methylxanthines levels were found to be associated with decreases in AO activities. 

In the fermentation of cupuassu, TPC, theobromine, and caffeine contents decreased by 60%, 70%, and 68%, respectively, while DPPH and ABTS activities decreased by 50% and 25%, respectively [94]. Fermentation acted similarly on cocoa beans by reducing TPC by 30–62% and DPPH by 25–82.5% [75,106]. To turn beans into chocolate, all steps of the process have an impact on the AO activity of the final product. For example, Bordiga et coll., showed that chocolate had a TPC and DPPH activity 2 and 12 times lower than cocoa beans, respectively [107]. Even if the final product had low AO activity, it is not negligible and may depend on its formulation. Indeed, milk and dark chocolates had lower DPPH activity (13.7 ± 1.2 and 75.1 ± 14.1 μmol TE/g of DW for cocoa and 4.5 ± 0.2 and 7.8 ± 0.4 μmol TE/g of DW for cupuassu) than their respective powder (239.4 ± 0.4 and 13.0 ± 0.5 μmol TE/g of DW for cocoa and cupuassu, respectively) [108,109]. 

An in vivo streptozotocin-induced diabetic rats model revealed that cocoa and cupuassu liquor consumptions increased the plasmatic AO capacity (ORAC and FRAP) and liver and kidney FRAP activities [85]. Moreover, AO enzymes could be deactivated in several factors of diabetes in the form of hyperglycemia, which causes oxidative stress [110]. Consumption of cocoa and cupuassu liquor increased AO enzyme activities such as kidney catalase (CAT), plasmatic SOD, and plasmatic GPx. Cocoa activated liver GPx and cupuassu activated brain CAT, SOD, and kidney GPx, but decreased hepatic SOD [85]. 

AO enzymes can scavenge ROS formed by lipid peroxidation and, by extension, protect cells from oxidative damages [111]. When comparing cocoa and cupuassu in an HFD animal model (that mimics human metabolic syndrome [112]), both increased plasmatic GPx activity, but also other AO enzymes such as plasmatic SOD and hepatic CAT activities. Cupuassu extract was the only one that decreased brain GPx activities [111]. Cupuassu and cocoa liquors induced a decrease in hepatic and plasmatic MDA levels and an increase in plasmatic AO abilities (FRAP and DPPH).

### 4.2. Immunomodulative (IM) Activities 

Inflammation is a complex protective multipathways process that is composed by the innate system (non-specific response) and the adaptative system (specific response). Although inflammation is a one-time response to eliminate a threat, it could become dangerous by becoming chronic, uncontrolled, and when directed against the body itself. Indeed, the organism could develop a hypersensitivity that could lead to activation of the complement, increased vascular permeability, platelet aggregation, and the secretion of enzymes by polynuclear neutrophiles that can lyse vessels and tissues [113,114]. Chronic inflammation could activate macrophages that release pro-inflammatory cytokines and cytotoxic lymphocytes [112] but also produce ROS and RNS (reactive nitrogen species) by membrane proteolytic enzymes called NADPH oxidase [115,116]. Cocoa and cupuassu presented anti-inflammatory activities by reducing ROS and RNS generation, but also interfered in the innate and adaptative immune systems. 

For the innate system, in addition to other pathways (microbial products or pro-inflammatory cytokines), ROS and RNS can cause the activation of the transcription factor NF-KB. It induces the proliferation of pro-inflammatory cytokines such as tumor necrosis factor alpha (TNF-α) or interleukins (IL-1α, IL-6 and IL-12) [117,118]. Macrophages from monocytes play a capital role in inflammation process. On the one hand, M1 macrophages contribute to inflammatory propagation by producing pro-inflammatory cytokines (IL-1β, IL-6, IL-12 and TNF-α), RNS, ROS, and by promoting T-helper 1 that produces interferon gamma (IFN-α) [117]. On the other hand, M2 macrophage is involved in wound healing and the resolution of chronic inflammation by excreting anti-inflammatory IL-10 and small amounts of IL-12 [117,119]. 

In a human THP-1 macrophages-derived model, cocoa extracts have shown IM activities by reducing proinflammatory cytokines (TNF-α, IL-6, IL-1β, and IL-12) and increasing anti-inflammatory cytokines (IL-10) in LPS/IFNγ-stimulated M1 [120]. Although cocoa did not affect M0 and M2 cytokines levels, they induced in M1 similar IL-10 and IL-12 levels as M2. This could suggest that cocoa may promote the polarization of M1 to an alternative M2 phenotype [120]. 

In a cell model using Mouse immortalized Mesangial Cells (MiMC), the addition of a high glucose (HG) level induced an increase in nitric oxide (NO) and ROS levels. Cupuassu extract reduced NO and ROS levels in MiMC treated with HG after 48 and 72 h [121]. In the in vivo diabetic rat model, cupuassu extract again decreased renal NO level, but also eNOS (endothelial nitric oxide synthase) and 3-nitrotyrosine (biomarker of RNS damages) levels from kidney. These results were also accompanied by a decrease in renal NF-KB and IL-6 suggesting the immunomodulatory potential of cupuassu [121]. In another in vivo intestinal inflammation rat model called TNBS (trinitrobenzenesulphonic acid)-induced rats, cupuassu not only decreased IL-6 and IL-1β levels, but also reduced the activity of mieloperoxidase (MPO, a toxic enzyme that produced ROS) and alkaline phosphatase (ALP, a marker of intestinal inflammation caused by lipid peroxidation) [122]. 

Although many studies presented the anti-inflammatory actions of cocoa, several indicated opposite actions. For example, Ramiro and collaborators indicated that cocoa extract decreased the TNF-α levels by 50–60% [123] while other studies indicated increases of 50% to 412% [75,124]

For adaptative system, in ovalbumin (OVA)-sensitized rat models, cocoa diets have been down-modulated OVA-specific antibodies IgM, IgG1, IgG2a and IgG2c [125] and IgE [126]. Allergen-specific T-cells turn into T helper 2 (Th2) and produce IL-4, responsible for producing IgE against the allergen [126]. The decrease in the secretion of IL-4, IgE, and IgG1 (main subclasse associated with Th2 immune response) after cocoa consumption suggested that it would act in Th2 immune down-response [125,126]. 

### 4.3. Impact on Intestinal Tract

As stated above, cocoa’s and cupuassu’s anti-inflammatory capacities have been demonstrated in various systems. Cocoa samples showed protective effects against intestinal damage on rats and cell models. Cocoa bean shell (extracted alone or incorporated in an ice cream formula) decreased the levels of two markers of inflammation, IL-8 and monocyte chemoattractant protein-1 (MCP-1), in Caco2-cells [127]. Similarly, the cocoa diet in Zucker diabetic fatty rats (ZDF) reduced the expression of TNF-α, IL-6, and MCP-1, but also the cd45 level (an immune marker of cell infiltration) [128]. Moreover, cocoa protected the integrity of the intestinal epithelial barrier by restoring protein levels involved in tight junctions (TJ). They included claudin-1, occluding, and anti-junctional adhesion molecule-A (JAM-A) [127], as well as Zonula occludens-1 (ZO-1) [128], which may be altered during intestinal damage. 

Metabolic disorder could lead to ROS production in hepatocytes and engender cell damage markers such as an increase in transaminase enzymes, for example alanine aminotransferase (ALT) and aspartate aminotransferase (AST). Consumption of cocoa and cupuassu liquor in HFD (High fat diet)-Winstar rats resulted in a reduction of these two markers. This indicated that they prevented mitochondrial hepatocyte damage caused by ROS production [85].

Accumulating various compounds in the digestive tract may result in local beneficial activities. For instance, ethanolic and aqueous extracts of cupuassu (bean and shell) were generated. Aqueous cupuassu bean extract exhibited a higher α-amylase inhibition rate (97.4 ± 1.01%) than other extracts (ranging from 57.3 ± 0.2 to 75.4 ± 0.9%) [129]. α-amylase had led to the starch cleavage increasing of glycemia. Cocoa has also shown antidiabetic effects by inhibiting these types of enzymes such as α-amylase and α-glucosidase [130,131,132].

A cocoa diet has induced changes in the microbiota composition. Indeed, when compared with non-diabetic ZDF rats who were not fed with cocoa, cocoa consumption led to an increase of the abundance of *Proteobacteria* (3.6-fold), *Tenericutes* (2.8-fold), and *Actinobacteria* (2.6-fold) phyla. Moreover, the cocoa diet significantly increased the abundance of *Firmicutes* (1.4-fold increase) and *Deferribacteres* phyla (9.3-fold increase) and decreased the relative abundance of *Cyanobacteries* phylum (by 74.9%) [128]. In human clinical trials, cocoa consumption has increased the abundance of *Blautia* and *Lachnospira* genera and decreased the abundance of *Agathobacter* genus and *Faecalibacterium prausnitzii* [133,134]. 

### 4.4. Clinical Trials of Cocoa 

Clinical trials of cocoa consumption revealed the potential of their bioactive compounds in multitude pathways. They would act in food allergy. This hypothesis was reinforced by a study of the relationship between health status and cocoa consumption. Authors found that the percentage of allergic students who are moderate and high consumers of cocoa was lower than that of the low consumers group. Moreover, cocoa consumption was associated with a lower allergic symptoms level [135]. Cocoa polyphenols could also affect the lymphoid composition of tissues where they were the most accumulated (thymus, lymph nodes and spleen) [136]. Indeed, in the rat model, a 10% cocoa-diet may have enhanced thymic maturation by reducing or increasing thymocytes (lymphocytes from thymus). A mature thymocyte exhibited CD4 and CD8 co-receptors on its surface with a T-cell receptor called TCRαβ high. The cocoa diet decreased the levels of CD8+, CD4- (and CD8-, CD4-) TCRαβ low (immature) cells without affecting the levels of those with TCRαβ high (mature). Moreover, the cocoa diet decreased CD8+ and CD4+ thymphocytes and increased the levels of CD8-, CD4-, and both CD4+ cells [137].

In a double-blind, placebo-controlled, randomized cross-over clinical trial, the effect of taking a single dose of high cocoa polyphenols was analyzed on redox statue and inflammatory response. The transcriptomic profile of the peripherical mononuclear cells (PBMC) of healthy volunteers revealed that after cocoa consumption, moderate modulation was observed and the gene expression was linked to (1) a decrease in ROS production (FPR1, IL8, Sestrin 3 (SESN3), CD36, and Hemoglobin Subunit Alpha 1/2 (HBA1/HBA2) (2) Ca^2+^ modulation (ADRB2, IL8, IL8RA, IL8RB, FPR1, Protein Tyrosine Phosphatase, Receptor Type C (PTPRC), TPT1 HBA1/ HBA2, ORM1), and (3) activation of leukocytes and viral response ((IL8, PTPRC, TIGIT, TPT1, FPR1, IL8RA and IL8RB). Of 98 genes with modulated expressions, 5 were selected for their involvement in at least these 3 regulatory pathways (IL8, IL8RB, CD36, ADRB2 and FPR1) and were validated by 40 RNA samples belonging to 10 participants in real time qPCR [138]. This hypothesis was reinforced by a double-blind, 12-week cocoa flavanol supplementation. Inflammation biomarkers (IL-6) and cardiometabolic risk factors seemed to be positively affected by this consumption [139]. This consumption also reduced levels of malondialdehyde (lipid peroxidation marker) and protein carbonylation. This indicated the implication of flavanols in blood oxidative stress diminution. Moreover, cocoa flavanol was associated with an improvement of mobility by being correlated with better hand strength, sit up, and walking distance assays [139]. The implications of cocoa for quality of life is increasingly being studied, while information about the cupuassu is still lacking. Their bioactive compounds were thought to be related to IM activity, so one might assume that with a very narrow composition, cupuassu could also have a significant impact on the immune system. More studies are required on this promising matrix. All pharmacological activities presented below are in Table 3.

## 5. Bioaccessibility and Bioavailability

Bioaccessibility could be defined as the fraction of a compound, released from the food matrix, capable of passing through the intestinal barrier, to reach the gut [140]. Meanwhile, bioavailability is the fraction of the available digested bioactive compound that can be taken through regular metabolic and distribution pathways [141]. Various parameters could affect the bioavailability of these compounds such as solubility, interaction with other food compounds, metabolism environment conditions, cell transporters, and interaction with the intestinal microbiota [142]. Furthermore, there are inter-individual parameters between consumers such as diet, genetic background, composition, and mechanism of intestinal microbiota [143]. Only 5–10% of the total intake of flavonoids (mainly monomeric and dimeric structures) can pass through the small intestine. This ability is often possible following deconjugation reactions such as deglycosylation. The remaining 90–95% can be converted into other metabolites with other implications by undergoing the gut microbiota fermentation process [141,144].

Cocoa polyphenols, absorbed in the small intestine, can reach the liver through the portal vein, where they can be converted by phase I and II biotransformations into new compounds [145]. In the liver, during phase I, compounds undergo oxidation, reduction, and hydrolysis. In phase II, they conjugate to became glucuronide, sulfated, and methyl derivates [146,147,148,149], and can reach various target organs. For example, once absorbed, (–)-epicatechin metabolites can reach lymphoid organs (thymus, spleen, etc.) and the liver [136]. Sulfate-glucuronide-(–)-epicatechin derivates have been identified in urine and plasma after ingesting dark chocolate [150].

Cocoa and cupuassu are rich sources of polyphenols. In the case of cocoa, it contains mainly flavanols such as (–)-epicatechin, (+)-catechin, and procyanidins (dimeric flavan-3-ol compounds such as B2). (–)-epicatechin can be rapidly absorbed in the small intestine (within 1 to 4 h). Conversely, large polyphenols such as procyanidins are poorly absorbed (10–100-fold less) [151,152,153]. Moreover, procyanidin and condensed tannins which are not allowed to pass through the gut barrier (due to their high weight structures or the nature of their sugar moiety) can be converted by the gut microbiota into various phenolic acids, to be absorbed [154]. 

Barros et coll., evaluated the distribution of cupuassu polyphenols in an animal model. They revealed that clovamide and flavanols ((−)-epicatechin and procyanidin B2) were present in higher doses in the stomach and small intestine than in the caecum and colon. An increase in (–)-epicatechin and B2 concentrations was observed and appeared to be related to the tannin hydrolysis process [74]. Flavones (hypolaetin glucuronide, hypolaetin glucuronide- sulphate, hypolaetin methyl ether and isoscutellarein) were found in the stomach, the small intestine, and in the caecum. In the colon, hypolaetin glucuronide was not detected, while higher amounts of hypolaetin glucuronide-sulfate were noted [76], likely due to enterohepatic circulation [155]. 

Microbial metabolism induces the presence of new compounds. For example, 1-(3,4-dihydroxyphenyl)-3-(2,4,6-trihydroxyphenyl) propan-2-ol and 1-(3-hydroxyphenyl)-3-(2,4,6-trihydroxyphenyl) propan-2-ol appeared to originate from (–)-epicatechin [76]. Some beverages made from cocoa beans, shells, and other ingredients (such as coconut, turmeric, and curry) were digested. They revealed differences in the amounts of methylxanthines and phenolic compounds. For example, for a coconut-cocoa beverage, after in vitro digestion, there was a decrease in phenolic acids (protocatechic acid, caffeic acid), flavanols (catechin, epicatechin and catechin-3-O-glucoside) and procyanidin B2. If the amounts of theobromine and caffeine remained unchanged after digestion, quercetin-3-O-glucoside and quercetin-3-O-rhamnoside were no longer detected [132].

When Andrade et coll., submitted their cupuassu (beans and shells) ethanolic and aqueous extracts to various digestive modalities (undigested, digested, and fermented), their composition revealed variations. This seemed to be influenced by the choice of the cupuassu part and the solvent, as well as the acidic and enzymatic state of the digestive compartment (leading to a difference in solubility). For example, gallic acid (from ethanolic beans extract) was detected after digestion and fermentation. Gallic acid was only detected after digestion of the shell. For the aqueous shell and bean extracts, it was only detected after the fermentation process [129]. The bioaccessibility index is defined by the concentration of compounds after ingestion over the concentration of the total compound. Regarding total polyphenol (TPC), digestion induced an increase in the bioaccessibility rate. Indeed, we noted an increase of 274 ± 4% and 203 ± 2%, for aqueous shell extract and ethanolic seed extract, respectively [131]. Dantas et coll., determined the bioaccessibility index by assessing the level of phenolic compounds that crossed the intestinal barrier (represented by a dialysis membrane), after in vitro digestion. They reported that catechin, procyanidin B1, B2 and A2 from cupuassu presented a bioaccessibility rate of 20, 62, 56 and 72%, respectively [156].

## 6. Perspectives

Both fruits are involved in the agri-food sectors but are valued differently. Cocoa seed is preferred over pulp, which is mainly used to initiate the bean fermentation process, while cupuassu pulp is directly transformed into dessert cream, sorbet and ice cream, juice and jam, and the seeds are less consumed [1]. This trend may change as more and more attention is paid to bean-based cupulat derivates. This could be a good alternative to chocolate, which is an increasingly challenging and problematic sector. For years, cocoa has been confronted with explosive demand (for beans and butter), which leads to ecological pressure. From 2000 to 2020, worldwide cocoa bean production rose from 3.38 million tonnes to 5.75 million tonnes, which represents $1.937 M and $7.404 M [157], respectively. 

This may result in issues of concern such as deforestation, climate change, and pollution. The cultivation of cocoa requires humid tropical regions, which limits opportunities for increased production. Other ways are being set up to produce more, such as the use of pesticides, over-consumption of water, and deforestation. Renier et coll, identified that over 2000–2019, there was 2.5 Mha of cocoa deforestation and degradation, which accounted for 46% of global deforestation [158]. Deforestation is a catalyst of global warming and limits cocoa cultivation by causing a water deficit due to evapotranspiration, modifying the cocoa growth condition, and causing a global temperature increase (2 °C by 2050) [159]. Thus, global warming causes an increase in drought in a huge water consumer culture. Santosa et coll., indicated that the changes in rainfall and temperatures (2010–2015) have been correlated with the fluctuation of cocoa production and could reduce population stability. Rainfall seemed to have a higher impact than temperature. However, other parameter could interfere, such as genotype and pedoclimatic conditions [160]. Cupuassu are also sensitive to water deficit. Like cocoa, it grows better in shaded areas and would be more productive when grown intercropped with other species [161]. It takes short- and long-term strategies to maintain cocoa production [160]. Cupuassu could be a good strategy to satisfy the producer with new growing areas and the consumer, who are looking for new products with traceability, ethics, and original flavor.

More and more studies are putting forward formulations based on cupuassu. Pereira et coll., indicated that cupuassu juice had a good probiotic carrier potential for *Lactobacillus casei* and their fermentation induced an increase in AO activities [162]. Costa et coll., formulated cupuassu prebiotic and probiotic goat milk yogurt [163]. However, with the rise of interest in superfoods and health products, consumers are not only looking for flavor, but also for a positive impact on their mental and physical health. Having an interesting composition is beneficial to health only if the compounds can be absorbed and metabolized by the human body. Cupuassu polyphenols (rhamnetin, gallic acid, epicatechin or pyrocatechol) were detected after digestion, but also (for vanillin) after fermentation by gut microbiota. Moreover, some compounds cannot reach the blood and instead target organs, and they can induce local activities in the intestinal tract [129]. However, there is limited evidence on the impact of this process on their composition and biological potential.

As the production of cocoa and cupuacu grows, so does the amount of plant waste. Pods of cupuassu and cocoa represent a burden for producers who must get rid of them, creating sanitary and social inconveniences. In 2019 and 2020, global exports of cocoa hulls, skins, and other waste amounted to $206 million and $225 million, respectively [164]. Côte d’Ivoire dominates this market with more than 80% of these exports over the two years. Most of this waste is disposed of even though it is a product with a high recovery potential. In the agri-food sector, new solutions are being created such as using cocoa pods as a source of sugaring molecules, for example as xylitol (by fermentation of the yeast *Candida boidinii XM02G*) [165], vegetable gum [100], and encapsulating agent [166], but also as an ingredient in the formulation of animal feed [167]. They can also interact in the technological part by being converted into biobased carbo-rich solids, such as active carbon (ref or replace fossil gas in certain thermal treatments (roasting, boiling water)). Zinla et coll. indicated that cocoa pods had thermochemical potential with a higher heating value of 13.70 MJ/kg. This heat provided by combustion could be used by thermal power plants to produce electricity. With the high level of potassium in cocoa bean ash (77.53%), it could also be used in the production of biofertilizers [168]. With a desire to be more environmentally friendly, the lack of information about the chemical composition of the cupuassu pod must also rectified.

## 7. Conclusions

In conclusion, cupuassu and cocoa fruits have varying compositions within and between species. Firstly, beans had higher lipid, polyphenol total, and methylxanthines contents than pods. Secondly, cupuassu seeds, which had similar levels of lipids and carbohydrates, contained fewer proteins, total polyphenols, and methylxanthine than cocoa. Both species may exhibit unique compounds that are not detected in the other. HPLC analysis revealed in cupuassu the presence of singular sulfated polyphenolic compounds named theograndin I and II and hypolaetin-derived, while cocoa have vitexin and theophylline. Cupuassu, which has a very similar composition, could be a good alternative to reduce the environmental pressure caused by the huge demand for cocoa. Both could be used in agri-food, cosmetic, and pharmaceutical applications in different formulations using beans and pods. These fruits are an interesting matrix due to their health potential. 

## Figures and Tables

**Figure 1 foods-11-03966-f001:**
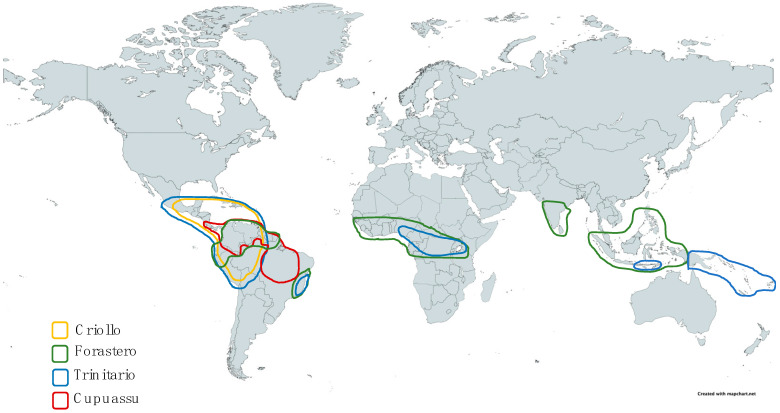
Approximative locations of *T. cacao* (Criollo, Forastero and Trinitario) and *T. grandiflorum* (Cupuassu). Data from [1,9,20,21,22,23,24].

**Figure 2 foods-11-03966-f002:**
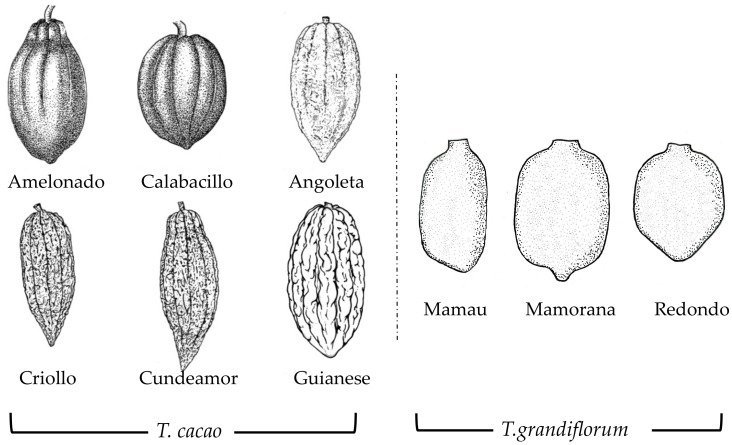
Representation of cocoa (adapted with permission of P. Lachenaud, Biotope Editions, 2022, [38]) and cupuassu pod shapes. Data from [2,38,39,40].

**Figure 3 foods-11-03966-f003:**
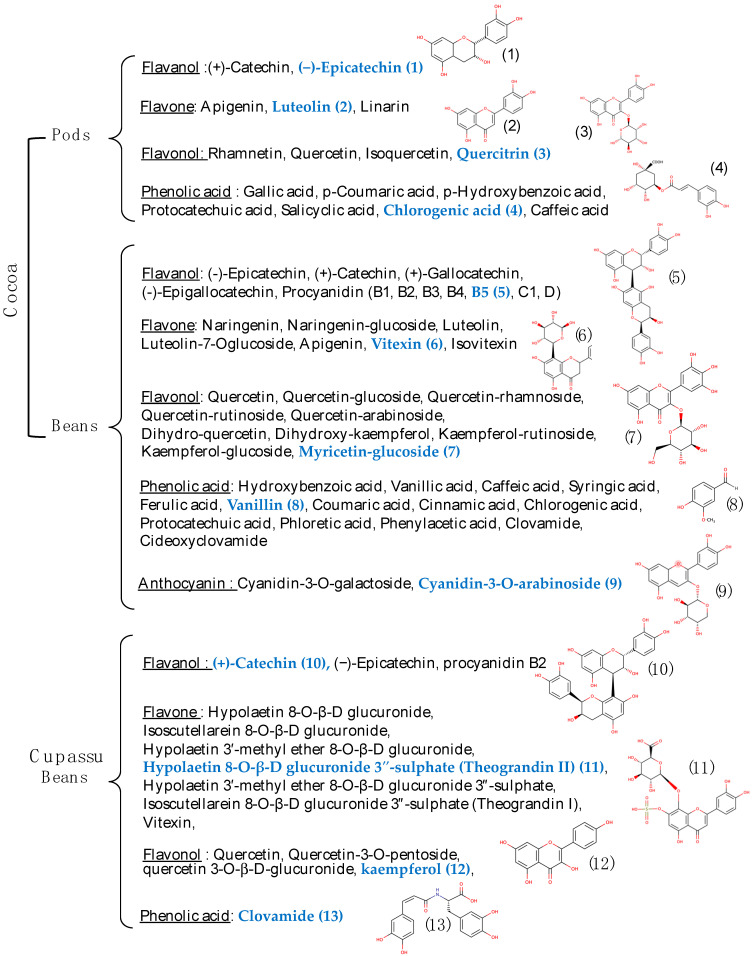
Polyphenols identified in pods and beans of cocoa and cupuassu, with examples of their chemical structures (in blue). Data from [9,50,52,69,70,71,72,73,74,75,76].

**Figure 4 foods-11-03966-f004:**
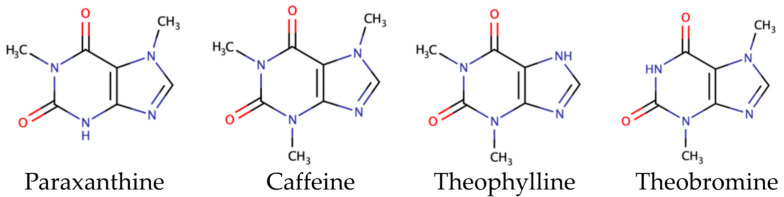
Methylxanthines structures.

**Table 1 foods-11-03966-t001:** Sections and species from the *Theobroma* genus and Sterculiaceae family. Data from [2].

Sections	Species
*Andropetalum*	*T. mammosum* Cuatr. & Leon
*Glossopetalum*	*T. angustifolium* Moçiño & Sesse, *T. canumanense* Pires & Fróes, *T. chocoense* Cuatr., *T. cirmolinae* Cuatr., *T. grandiflorum* (Willd. ex Spreng.) Schum., *T. hylaeum* Cuatr., *T. nemorale* Cuatr., *T. obovatum* Klotzsch ex Bernoulli, *T. simiarum* Donn. Smith., *T. sinuosum* Pavón ex Hubber, *T. stipulatum* Cuatr., *T. subincanum* Mart
*Oreanthes:*	*T. bernouillii* Pittier, *T. glaucum* Karst, *T. speciosum* Willd., *T. sylvestre* Mart, *T. velutinum* Benoist
*Rhytidocarpus*	*T. bicolor* Humb. & Bonpl.
*Telmatocarpus*	*T. gileri* Cuatr., *T. microcarpum* Mart.
*Theobroma*	*T. cacao* L.

**Table 2 foods-11-03966-t002:** Comparison of the chemical composition of cupuassu and cocoa.

	Beans		Pods	
	Cupuassu	Cocoa	Cupuassu	Cocoa
Proteins (g/100 g dry weight)	2.2–26.2 [51,52,53,55]	2.5–14.4 [54,56,57,58]	-	2.4–17.6 [45,46,47,48,49,50]
Lipids (g/100 g dry weight)	20–60 [42,51,53]	20–60 [54]	-	0.6–2.3 [45,46,48,49,50]
Total sugars (g/100 g dry weight)	1.3–1.6 [66]	0.1–3.1 [67]	-	1.7 [45]
Polyphenols (TPC-mg AG eq./g dry weight)	7.8 [85]	9.8–202.2 [75,79,80]	-	3.2–56.5 [46,69,77,78]
Caffeine (g/100 g dry weight)	0.05 [68]	0.2–2.1 [1,51,83,88,89,90,91]	-	0.002–0.004 [1,51]
Theobromine (g/100 g dry weight)	0.1 [68]	0.2–7.6 [1,51,83,88,89,90,91]	-	0.002–0.4 [1,51]

**Table 3 foods-11-03966-t003:** Summary of pharmacological activities of cocoa and cupuassu.

Antioxydant activities	**Model**	**Matrix**	**Action**	**Sources**
	Cell model-human EA. Hy926 and human HepG2	Cocoa	↓ MDA level, ↓ ROS level	[97,98]
			↑ GPx and GR activities	
	Caco-2 epithelial cell and murine STC-1 enteroendocrine cells	Cupuassu	↓ ROS levels	[99]
	Rats model	Cupuassu	↓ ROS levels	
	Ewes model	Cocoa	Not affect AO plasmatic activities,	[101]
			↑ SOD activities,	
			↓ protein carbonyl levels	
	HFD animal model	Cocoa and Cupuassu	↑ plasmatic GPx, SOD and hepatic CAT activities	[111]
			↓hepatic and plasmatic MDA levels	
			↑FRAP and DPPH plasmatic activities	
		Cupuassu	↓ brain GPx activities	
	STZ-induced diabete rats model	Cocoa and Cupuassu	↑CAT kidney, SOD plasma and plasmatic GPx	[85]
			↑ plasmatic, liver and kidney FRAP and plasmatic ORAC activities	
		Cocoa	↑ liver GPx activity	
		Cupuassu	↑ bain CAT and SOD,	
			↑ kidney GPx	
Immunomodulatory activities	human THP-1 macrophages M1 model	Cocoa	↓TNF-α, IL-6, IL-1β, and IL-12 levels,	[120]
			↑ IL-10 level,	
			Provide M1/M2 metabolic switch (similar levels of IL-10 and IL-12)	
	MiMC cell model	Cupuassu	↓ NO and ROS levels,	[121]
	in vivo diabetic rat model	Cupuassu	↓ renal NO level, kidney eNOS and 3-nitrotyrosine,	
			↓ renal NF-KB and IL-6 levels	
	in vivo TNBS-induced rats model	Cupuassu	↓ neutrophil myeloperoxidase and alkaline phosphatase activities,	[122]
			↓ IL-6 and IL-1 levels	
	ovalbumin-sensitized rats models	Cocoa	↓ OVA-specific antibodies IgM, IgG1, IgG2a and IgG2c and IgE levels	[125,126]
	Rat model	Cocoa	↓ the levels of CD8+, CD4- (and CD8-, CD4-) TCRa-b low (immature) cells,	[137]
			↓ CD8+, CD4+ thymphocytes	
			↑ CD8-, CD4- and both CD4+ cells levels	
Impact on intestinal tract	Caco2 cells	Cocoa	↓ IL8 and MCP1 levels, restoring claudin-1, occludin and JAM-1 levels	[127]
	ZDF rats	Cocoa	↓ TNF-α, IL-6 and MCP1 levels, and ↓ cd45 levels	[136]
			restauring ZO-1	
	HFD rats	Cocoa	↓ ALT and AST activities	[107]
	ZDF rats	Cocoa	↑ *Proteobacteria, Tenericutes* and *Actinobacteria* phyla,	[128]
			↑ *firmicutes* and *deferribacteres* phyla,	
			↓ *cyanobacteries* phylum	
Clinical trials	Human	Cocoa	↑ *Blautia* and *Lachnospira* genera,	[133,134]
			↓ *Agathobacter* genus and *Faecalibacterium prausnitzii*	
		Cocoa	modulated the expression of gene involving in ↓ of ROS production, Ca^2+^ modulation and activation of leukocytes and viral response.	[138]
		Cocoa	↓ IL6, malondialdehyde and protein carbonyl levels, ↑ 6-min walked distance assay, ↓ sit-up test, ↑ hand strength assay	[139]

The increase and decrease are represented by ↑ and ↓ respectively in the table.

## Data Availability

No new data were created or analyzed in this study. Data sharing is not applicable to this article.

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
