# Peer review of "Theobroma cacao and Theobroma grandiflorum: Botany, Composition and Pharmacological Activities of Pods and Seeds"

_foods, 2022, doi:10.3390/foods11243966_

Round 1
Reviewer 1 Report
The manuscript entitled "Theobroma cacao and Theobroma grandiflorum: Botany, Com- 2 position and Pharmacological activities of pods and seeds" presents interesting information related to the taxonomy and origins, bioactive compounds and potential pharmacologial potencial go the cacao y cupuassu beans and pods. the work presents interesting information however, some minor recommendations are needed:
make a table to include the section and species of theobrma
make a table of the chemical composition of beans and pods to easily identify their differences
Author Response
The manuscript entitled "Theobroma cacao and Theobroma grandiflorum: Botany, Composition and Pharmacological activities of pods and seeds" presents interesting information related to the taxonomy and origins, bioactive compounds and potential pharmacologial potencial go the cacao y cupuassu beans and pods. the work presents interesting information however, some minor recommendations are needed:
Response: Thank you very much for your comments which helped us improve this manuscript. All the document has submitted an English revision.
- make a table to include the section and species of theobrma
Response : Thank you for your advice, we included these information on a table called Table 1
- make a table of the chemical composition of beans and pods to easily identify their differences
Response : Thank you for your suggestion, this table was added to the document.
Reviewer 2 Report
In lines 48 - 50 you say that cupuassu can be found in Venezuela and Colombia; however, in figure 1 I can't see that. Why you don't show in this figure location of cupuassu in these countries?
Why there aren't information related to agro ecological requirement for cupuassu? Are there the same that cocoa?
Which are the specificities for Guiana? I suggest to say it.
How is made polinization for cupuassu? Is self - compatible, incompatible or self-incompatible? What do you know about it?
If in Brazil there are three cultivars of cupuassu are well known: Redondo (with a rounded apex), Mamorano (with a pointed apex) and Mamau (a hypotetic parthenocarpic mutant), why there are only one shape representation for cupuassu? This representation is for Redondo? Which are the representation for Mamorano y Mamay?
I suggest you using the same name: husk or pod.
In lines 186 – 188 I don't undestood why do you say 44.6% of essential amino acids, but individual amino acids is not mayor than 4.65. In addition you use , and . for decimals.
In lines 194 – 196 Please verify %, because is greater than 100%.
In line 244 TPC It's depending of cocoa's genotype. I suggest to include the follows reference:
S. M. Pico-Hernández, C. J. Murillo-Méndez, and L. J. López-Giraldo, “Extraction, separation, and evaluation of antioxidant effect of the different fractions of polyphenols from cocoa beans,” Rev. Colomb. Quim., vol. 49, no. 3, 2020, doi: 10.15446/rcq.v49n3.84082.
Line 255 this cocoa's bean is fermented and dried? I think that's dried!
For antioxidant and antinflamatory activities your discussion is based on extract from cocoa's beans, powder and liquors (only one study using COP was used for discussion) how can you guarantee that behavior for COP will be the same? In my opinion behavior will be different as results of polyphenol profile of cocoa's extract and COP extract (table 1)
Finally, in my opinion the discussion related to pharmacological activity of pods is poorly sustained, may be there aren’t enough information about it; therefore, I suggest you to include this fact in your discussion.
Author Response
Response: Thank you very much for your comments which helped us improve this manuscript. All the document have submitted an English revision.
- In lines 48 - 50 you say that cupuassu can be found in Venezuela and Colombia; however, in figure 1 I can't see that. Why you don't show in this figure location of cupuassu in these countries?
Response : Thanks for your comment. We added these elements in the Figure 1.
- Why there aren't information related to agro ecological requirement for cupuassu? Are there the same that cocoa?
Response : Cupuassu and cocoa have compatible agroecological requierment. We have included some elements in this section.
- Which are the specificities for Guiana? I suggest to say it.
Response : Thanks for the reminding, we included the missing information about guiana flowers.
- How is made polinization for cupuassu? Is self - compatible, incompatible or self-incompatible? What do you know about it?
Response : Theobroma species had a low fertilization capacity. To clarify the content we have indicated that the two species are both self-incompatible. Cupuassu would behave like cocoa. Ramos et al, (2005) suggested that incompatibility between gametes would appear after fertilization and would be due to a delay in the pollen tube and an inhibition in the ovary before fertilization.
- If in Brazil there are three cultivars of cupuassu are well known: Redondo (with a rounded apex), Mamorano (with a pointed apex) and Mamau (a hypotetic parthenocarpic mutant), why there are only one shape representation for cupuassu? This representation is for Redondo? Which are the representation for Mamorano y Mamay?
Response : It is very hard to find a representation of the shapes of cupuassu. We have nevertheless modified the figure by representing these three characteristic types.
- I suggest you using the same name: husk or pod.
Response : It’s done. Thanks for this advice.
- In lines 186 – 188 I don't undestood why do you say 44.6% of essential amino acids, but individual amino acids is not mayor than 4.65. In addition you use , and . for decimals.
Response : We tried to clarify the sentence. The total mino acid content represent 11.6% of the cocoa pod sample, with 5.17% of essential amino acids. Thus, they accounted for 44.57% of the total amino acid. Attention paid to punctuation of decimals.
- In lines 194 – 196 Please verify %, because is greater than 100%.
Response : We corrected this part of the results.
- In line 244 TPC It's depending of cocoa's genotype. I suggest to include the follows reference: S. M. Pico-Hernández, C. J. Murillo-Méndez, and L. J. López-Giraldo, “Extraction, separation, and evaluation of antioxidant effect of the different fractions of polyphenols from cocoa beans,” Rev. Colomb. Quim., vol. 49, no. 3, 2020, doi: 10.15446/rcq.v49n3.84082.
Response :We have modified the section to highlight this information. Thanks you for this interesting reference.
- Line 255 this cocoa's bean is fermented and dried? I think that's dried!
Response : Theses beans were also fermented and dried. We clarified this information in the sentence.
- For antioxidant and antinflamatory activities your discussion is based on extract from cocoa's beans, powder and liquors (only one study using COP was used for discussion) how can you guarantee that behavior for COP will be the same? In my opinion behavior will be different as results of polyphenol profile of cocoa's extract and COP extract (table 1)
Response : You’re absolutely right, there is a cruel lack of information about antioxidant and anti-inflammatory activities of pods in the literature. This makes it difficult to compare them with cocoa beans, which are more studied. The purpose of this article is to highlight the lack of information on this matrix and to provide a willingness to find answers. When we compile data on AO activities, a trend seems to be emerging, but further studies are needed to reinforce this idea. This lower COP activity may be due to their polyphenol composition, which is less rich than beans or to the different nature of their compounds. Thanks for your comment. We have discussed it and have nuanced the results.
- Finally, in my opinion the discussion related to pharmacological activity of pods is poorly sustained, may be there aren’t enough information about it; therefore, I suggest you to include this fact in your discussion.
Response : Thank you. We include this fact in this section.
Reviewer 3 Report
Major corrections
- There are many published review articles about cacao. The authors should indicate the originality of the present manuscript.
- English grammar and editing are highly recommended in some sections of the manuscript.
- There are several studies on the beneficial effects of cocoa on humans. These clinical trials should be considered in a new section of the manuscript. This new section could improve the presentation and content of the review for readers.
- A perspective section should be included. What is the future relevance of these plant species for human health?
- Brief ecological information about these tree species should be discussed. Are their population stable?
- The conclusion section presents information previously mentioned in the above sections.
Minor corrections
- Some sentences are repeated (i.e., "Cocoa and cupuassu belong to the genus Theobroma and the family Malvaceae" in different sections of the manuscript).
- Figure 3. The authors are encouraged to draw the chemical structures of the compounds. They can use ChemDraw or other software.
Author Response
Major corrections
- There are many published review articles about cacao. The authors should indicate the originality of the present manuscript.
Response : Thank you for your guidance. We've clarified that in the introduction and in some of the discussion.
- English grammar and editing are highly recommended in some sections of the manuscript.
Response : Thank you very much for your comments which helped us improve this manuscript. All the document has submitted an English revision.
- There are several studies on the beneficial effects of cocoa on humans. These clinical trials should be considered in a new section of the manuscript. This new section could improve the presentation and content of the review for readers.
Response : We include a new section with clinical trials only.
- A perspective section should be included. What is the future relevance of these plant species for human health?
Response : We took your advice into consideration and included a "perspectives" section.
- Brief ecological information about these tree species should be discussed. Are their population stable?
Response : Some ecological data were added in the document.
- The conclusion section presents information previously mentioned in the above sections.
Response : Thanks for your comment, we modified the conclusion section.
Minor corrections
- Some sentences are repeated (i.e., "Cocoa and cupuassu belong to the genus Theobroma and the family Malvaceae" in different sections of the manuscript).
Response : Thanks for your attention, we deleted the repeated sentence.
- Figure 3. The authors are encouraged to draw the chemical structures of the compounds. They can use ChemDraw or other software.
Response : We modified Figure 3 by drawing the structure ourselves
Round 2
Reviewer 3 Report
The manuscript can be accepted for its publication in Foods